# Copy Number Variants in Cardiac Channelopathies: Still a Missed Part in Routine Arrhythmic Diagnostics

**DOI:** 10.3390/biom14111450

**Published:** 2024-11-15

**Authors:** Maria Gnazzo, Giovanni Parlapiano, Francesca Di Lorenzo, Daniele Perrino, Silvia Genovese, Valentina Lanari, Daniela Righi, Federica Calì, Massimo Stefano Silvetti, Elena Falcone, Alessia Bauleo, Fabrizio Drago, Antonio Novelli, Anwar Baban

**Affiliations:** 1Laboratory of Medical Genetics, Translational Cytogenomics Research Unit, Bambino Gesù Children’s Hospital, IRCCS, 00165 Rome, Italy; maria.gnazzo@opbg.net (M.G.); daniele.perrino@opbg.net (D.P.); silvia.genovese@opbg.net (S.G.); valentina.lanari@opbg.net (V.L.); antonio.novelli@opbg.net (A.N.); 2Cardiogenetic Center, Rare Diseases and Medical Genetics Units, Bambino Gesù Children’s Hospital, IRCCS, 00165 Rome, Italy; giovanni.parlapiano@opbg.net (G.P.); francesca1.dilorenzo@opbg.net (F.D.L.); 3Pediatric Cardiology and Arrhythmia/Syncope Complex Unit, Bambino Gesù Children’s Hospital, IRCCS, 00165 Rome, Italy; daniela.righi@opbg.net (D.R.); federica1.cali@opbg.net (F.C.); mstefano.silvetti@opbg.net (M.S.S.); fabrizio.drago@opbg.net (F.D.); 4Biogenet, Medical and Forensic Genetics Laboratory, 87100 Cosenza, Italy; biogenetsrl@gmail.com (E.F.); info@biogenet.it (A.B.); 5The European Reference Network for Rare, Low Prevalence and Complex Diseases of the Heart (ERN GUARD-Heart), 1105 Amsterdam, The Netherlands

**Keywords:** channelopathies, copy number variants, long QT, Brugada, catecholaminergic polymorphic ventricular tachycardia

## Abstract

Inherited cardiac channelopathies are major causes of sudden cardiac death (SCD) in young people. Genetic testing is focused on the identification of single-nucleotide variants (SNVs) by Next-Generation Sequencing (NGS). However, genetically elusive cases can carry copy number variants (CNVs), which need specific detection tools. We underlie the utility of identifying CNVs by investigating the literature data and internally analyzing cohorts with CNVs in *KCNQ1*, *KCNH2*, *SCN5A,* and *RYR2*. CNVs were reported in 119 patients from the literature and 21 from our cohort. Young patients with CNVs in *KCNQ1* show a Long QT (LQT) phenotype > 480 ms and a higher frequency of syncope. None of them had SCD. All patients with CNV in *KCNH2* had a positive phenotype for QT > 480 ms. CNVs in *SCN5A* were represented by the Brugada pattern, with major cardiac events mainly in males. Conversely, adult females show more supraventricular arrhythmias. *RYR2*-exon3 deletion showed a broader phenotype, including left ventricular non-compaction (LVNC) and catecholaminergic polymorphic ventricular tachycardia (CPVT). Pediatric patients showed atrial arrhythmias and paroxysmal atrial fibrillation. Relatively higher syncope and SCA were observed in young females. The detection of CNVs can be of greater yield in two groups: familial channelopathies and patients with suspected Jervell and Lange-Nielsen syndrome or CPVT. The limited number of reported individuals makes it mandatory for multicentric studies to give future conclusive results.

## 1. Introduction

Inherited cardiac arrhythmias (ICAs) are a heterogeneous group of genetic disorders characterized by an increased risk in affected individuals for life-threatening arrhythmias and SCD [1]. Particularly, they contribute greatly to morbidity and mortality rates in young individuals, representing the main cause of SCD in populations less than 35 years old, ranging from 27% to 40% [2]. Due to their clinical and genetic heterogeneity, the molecular diagnosis of these diseases may be challenging and almost never has a 100% detection rate [3]. Nevertheless, pinpointing a genetic cause can be crucial for both the prognosis and clinical management of affected individuals [4]. Nowadays, genetic testing is focused on the application of Next-Generation Sequencing (NGS) technology to identify single-nucleotide variants (SNVs) in genes encoding mainly ion channels leading to channelopathies [4]. Routine NGS technology, unless using dedicated pipeline or genome-wide techniques, is often blind in detecting CNVs [5]. The latter, increasingly recognized in ICAs, may represent an underestimated genetic cause of ICAs [6,7]. CNVs can be investigated through different techniques, including Multiplex Ligation-dependent Probe Amplification (MLPA) and chromosome microarray analysis (CMA); however, they are not largely used in the clinical routine of arrhythmias. There are several reports of pathogenic CNVs in individuals affected by channelopathies, especially involving genes associated with Long QT syndrome (LQTS) [8,9,10,11,12,13,14,15], Jervell and Lange-Nielsen syndrome (JLNS) [16,17,18], Brugada syndrome (BrS) [19,20,21,22,23,24], and catecholaminergic polymorphic ventricular tachycardia (CPVT) [25,26,27,28,29,30,31,32,33,34], with an overall estimated detection rate of 5% [6]. However, their real contribution has not been systematically studied so far.

The aim of this study is to underlie the utility of identifying CNVs in cardiac channelopathies after genetically elusive NGS studies. The identification of this underdiagnosed genetic cause will contribute to increased diagnostic yield, better clinical management of affected individuals, and preventive measures to avoid life-threatening events. This study includes a comprehensive literature survey (119 patients), supported by our cohort of 21 patients.

## 2. Materials and Methods

This study is divided into two major parts: literature data collection and revisited data of patients described with ICAs due to channelopathy genes CNVs, and the second group includes the same category of patients from our internal cohort after genetically elusive NGS studies.

We performed a survey of previous studies documenting CNVs in any genes of *KCNQ1*, *KCNH2*, *SCN5A*, and *RYR2*. We searched PubMed for published studies without restriction on the date of publication and without restrictions on language, using the search terms “channelopathies AND MLPA”; “channelopathies AND CNV”; “long QT AND CNV”; “brugada AND CNV”; “CPVT AND CNV”; “*RYR2* AND CNV”; “long QT AND MLPA”; “brugada AND MLPA”; “*SCN5A* AND MLPA”; “*KCNQ1* AND MLPA”; “*KCNH2* AND MLPA”; “*RYR2* AND MLPA” (accessed on June 2024).

Review and cohort studies with the above-mentioned specificity were included. The papers were carefully read and reconsidered accordingly. Two investigators performed the search independently. The references of the selected papers were crosschecked with the same inclusion condition. Duplicates were removed (Figure 1).

Regarding the 21 individuals analyzed at our center, informed consent for the genomic analyses and clinical data collection was obtained from the patients’ parents.

Genomic DNA was extracted from circulating leukocytes using the QIAampH DNA Blood Kit (QIAGEN Sciences, Germantown, MD, USA). Clinical Exome Sequencing (CES) was performed using ClinEX pro Kit (4bases, Manno, Switzerland) according to the manufacturer’s protocol and was sequenced on the Illumina NovaSeq6000 platform (Illumina, San Diego, CA, USA) to exclude the presence of pathogenic or likely pathogenic SNVs associated with ICAs.

The patients with inconclusive NGS testing were analyzed to investigate CNVs in cardiac genes through Multiplex Ligation-dependent Probe Amplification (MLPA) kits, according to the MRC-Holland protocol (MRC-Holland, Amsterdam, The Netherlands). MLPA was performed on genomic DNA using the SALSA MLPA P114-C1 mix, which includes probes for detecting deletions or duplications in *KCNQ1*, *KCNH2*, *KCNE1*, *KCNE2,* and *KCNJ2* genes, as well as SALSA MLPA P108-B4 with probes in the *SCN5A* gene.

The amplification products were separated by capillary electrophoresis using ABI Prism 3500xl Genetic Analyzer (Applied Biosystem, Waltham, MA, USA), and the results were analyzed using Coffalyser.Net v.240129.1959 software (MRC-Holland). Each proband probe peak was compared with reference probe peaks (intrasample normalization). By comparing the sample’s relative probe peak to all control samples, final probe ratios were calculated (intersample normalization).

Real-time polymerase chain reaction was used to validate the MLPA findings. Deletion of exon 3 of *RYR2* was detected by Chromosomal Microarray Analysis (CMA), performed using Infinium CytoSNP-850K BeadChip (Illumina, San Diego, CA, USA), according to the manufacturer’s protocol. Array scanning data were generated on the Illumina NextSeq 550 system, and the results were analyzed by the BluefuseMulti 4.4 software.

This study was approved by the institutional scientific board of Bambino Gesù Children’s Hospital and conducted in accordance with the Helsinki Declaration. All data were obtained in agreement with Bambino Gesù Children’s Hospital’s ethical standards.

In this study, we propose a practical flow chart reporting our method leading to the identification of the cohort with channelopathy genes (Figure 2).

## 3. Results

The results of both the literature survey and the details from our cohort are summarized in Table 1, including subcategories for the following genes: *KCNQ1*, *KCNH2, SCN5A,* and *RYR2*. The main gathered data included age, gender, and QTc interval when reported, and they are divided into three subgroups: QTc group 3 when >480 ms; QTc group 2 when 460–480 ms; and QTc group 1 when <460 ms, syncope, trigger, SCA/D, treatment options, and other significant clinical data.

The literature data include 119 individuals with CNV in one of the analyzed genes. Our cohort included 485 MLPA tests for patients without pathogenic or likely pathogenic variants in NGS analysis for cardiac and arrhythmic genes. We identified microrearrangements in 21 individuals (4.3%), and their detailed clinical features and family pedigrees are reported in Figure 3. The technical results of MLPA analysis are summarized in Figure 4.

The following paragraphs include, in a detailed manner, the main results from each subgroup.

### 3.1. KCNQ1 Related CNV

The total number of patients was 35. We opted purposely to exclude four, since they lacked all major clinical details with subsequent bias in data analysis. The cohort was divided into two groups on the basis of reported age (under or over 18 years old).


**The age at diagnosis was under 18 years old in 16 patients:**
Gender distribution included 11 females (68.75%) and 5 males (31.25%).QTc interval was >480 ms in 10 patients (62.5%), QTc 460–480 ms in 5 patients (31.25%), and it was normal in 1 individual (6.25%).Syncope was reported in 5 patients (31.25%) with a major triggering factor related to exercise. Two patients (12.5%) had seizures.Thirteen patients were on Beta-blocker (BB) treatment, and these data were missing for three patients. Implantable Cardioverter Defibrillator (ICD) was undertaken in two patients.



**The age at diagnosis was over 18 years old in 15 patients:**
Gender distribution included 12 females (80%) and 3 males (20%).QTc interval was reported in 12 patients: QTc > 480 ms in 5 patients (33.3%), QTc 460–480 ms in patients 4 (26.6%), and normal QT in 4 individuals (26.6%), and the data were missing for 2 patients.Syncope was reported in three patients (20%), while SCD was reported in one patient (6.6%). No major other clinical manifestations were described.Treatment with BB was reported in seven patients (46.6%), and these data were missing in eight.


JLNS was diagnosed in five patients (three children and two adults). Two patients carried homozygous CNV deletion, while the other three were compound heterozygous (CNV and SNV).

### 3.2. KCNH2 Related CNV

This category included 25 patients. Clinical data were missing in nine patients. The following results were calculated on the basis of 16 patients:LQTS: QTc > 480 ms in 13 patients (81.25%), QTc 460–480 ms in 3 patients (18.75%), and none were reported to have normal values.Syncope was represented in six patients (24%), of whom two were reported with SCD. Triggering factor for syncope was present in four individuals (three acoustic and one exercise). Epilepsy was reported in two individuals from the same family.The treatment section was difficult to develop in this cohort since it was mentioned in only six patients (five BB and one with ICD).

### 3.3. SCN5A Related CNV

The total number of patients was 28. The cohort was divided into two groups on the basis of reported age (under or over 18 years old).


**The age at diagnosis was under 18 years old in 14 patients:**
Gender distribution included seven females (50%) and seven males. The youngest reported patient was 8 years old.The cardiac phenotype was normal in three female patients. The pattern of BrS was observed in nine patients with variable supraventricular and ventricular electrical changes. Conduction system abnormalities were noticed in three male patients with Sick Sinus Syndrome (SSS) (in two of them, they showed BrS as well). Atrioventricular Block (AVB) was observed in three patients (two males and one female). Atrial flutter (AF), a rare event in the pediatric age group, was present in two male patients with BrS.Two patients showed syncope: one male who eventually received ICD and one female who had syncope secondary to pain as a triggering factor. One female patient had a major event of SCA during sleep and carried, in addition to the CNV, an SNV classified as a variant of unknown significance on *SCN5A*: c.3157G > A (p.Glu1053Lys).Treatment options included ICD in three patients (two males and one female) and a pacemaker (PM) in one male individual.



**The age at diagnosis was over 18 years old in 14 patients:**
Gender distribution included five females (35.7%) and nine males (65.3%).Two females had normal cardiac phenotype, and the other three had major supraventricular manifestations, including one patient with bradycardia, the second with SSS, AF, and BrS, and AVB in the third one. The males showed a wide spectrum, consisting in a pattern of BrS in eight patients with variable supraventricular and ventricular changes: AVB was observed in one patient (with Brs and ventricular tachycardia (VT)), and one patient showed Ventricular Fibrillation (VF) in Brs.Major events were observed only in five males, including four syncopes and three SCAs (two individuals showed combined syncope and SCA).Treatment mainly included ICD in five male patients.


### 3.4. RYR2 Related CNV

The total number of patients reported in the literature, in addition to our cohort, includes 52 individuals. This cohort was divided into two main groups: 42 with heterozygous exon 3 deletion, and 10 patients, described in a single report, carrying exon 1 to 4 duplication in homozygous.

The cohort of the 42 patients carrying *RYR2* exon 3 deletion was further subdivided into three groups: those under 18, those over 18 years old, and the group where age was not determined.


**Patients younger than 18 years old totaled 12:**
Gender distribution included seven females and five males.The clinical spectrum showed one SCA in a female (preceded by syncope), three females with syncope, and only one male with syncope; none of the males had SCA. In this cohort, four patients received ICD (three females and one male) and one PM (female). Bradyarrhythmias were divided into those with AVB (three), SSS/Sinoatrial Dysfunction (SAD) (four), and bradycardia (BC) in four patients. Atrial arrhythmias, which are rare events in the pediatric age group, were demonstrated in four patients. CPVT diagnosis was delivered in eight patients, and Ventricular Arrhythmias (VAs) were observed in three patients. Myocardial involvement was overrepresented in this group, with LVNC (left ventricular non-compaction) in eight patients, including Dilated Cardiomyopathy (DCMP) in two.



**Patients older than 18 years old totaled 22:**
Gender distribution included 12 females and 10 males.Events in females included one SCA (preceded by syncope) and four females with syncope. Two males reported syncope, but none had SCA. Five patients received ICD (four females and one male) and six PM (three females and three males). Bradyarrhythmias were divided into those with AVB (four), SSS/SAD (nine), and BC in five patients. Atrial arrhythmias were demonstrated in nine patients. CPVT diagnosis was reached in 10 patients, and Ventricular Arrhythmias were observed in 9 patients. Myocardial involvement was overrepresented in this group, with LVNC in seven patients, including DCMP in three.


The cohort of patients carrying a homozygous duplication, involving approximately 26,000 base pairs of intergenic sequence, *RYR2*’s 5′UTR/promoter region, and exons 1 through 4 of *RYR2*, included 10 individuals described in a single report, including eight females and two males. SCA was present in five females (four with reported age under 18 years old) and two males (unreported age).

## 4. Discussion

Given the severity of ICAs with the consequent risk of SCD and rapidly advancing therapeutic options, prompt diagnosis of probands and testing of family members is imperative [1,35,36]. Genetic testing in cardiac arrhythmias has been widely used [3,4]. In the last decade, NGS sequencing has represented the “gold standard” for the identification of the genetic basis of ICAs. However, the detection rate, in spite of confirmed clinical diagnosis, remains under 100% for the majority of these conditions. This leads to the presence of a genetically elusive group of patients who might be undiagnosed or misdiagnosed. In addition to these individuals, family members remain in the dark and unaware of whether they have an increased risk for adverse cardiac events (familial variable expressivity is the rule in ICAs) [37].

There is a progressively increasing knowledge that other laboratory tools/pipelines are able to integrate/increase NGS genetic yield. One of those is related to copy number variants (CNVs). The latter represent DNA rearrangements that some NGS platforms are blind to detect. This group of undiagnosed patients is decreasing since NGS methods in the last few years have become more and more able to detect CNVs in addition to SNVs.

The investigation of the literature and our experience allowed us to highlight 119 and 21 patients (4.3% of our patients with negative NGS testing), respectively, with ICAs caused by CNVs in major channelopathy genes.

From a clinical perspective, patients under 18 years old with CNVs in *KCNQ1* show a more severe LQT phenotype (62.5% vs. 33.3%) and a higher frequency of syncope (31.25% vs. 13.3%) compared with adults. However, none of those under 18 years old had SCD. It is important to mention that no precise statistical significance can be established due to the limited cohort and the need for more multicentric observation related to LQTS1 in *KCNQ1* CNVs.

In contrast, almost all patients with CNV in *KCNH2* had a positive phenotype for Long QT with a high percentage of QT > 480 ms (81.25%). This aspect is in line with the relative severity of the phenotype related to *KCNH2* channelopathies [8,9].

CNVs in *SCN5A* were represented by deletions. The most frequently associated phenotype was BrS in accordance with the loss of function mechanism related to this gene [15,19,21,23,30,38]. Brugada pattern and major cardiac events such as syncope and SCD are more evident in male individuals compared with females, who require life-saving devices. Conversely, adult females show more supraventricular arrhythmias.

In the last category, deletion of *RYR2* exon 3 resulted in altered protein structure and interaction with other *RYR2* domains. The phenotypic manifestation was broad both at the myocardial and arrhythmic level, including LVNC and CPVT [25,26,28,33]. Pediatric patients with *RYR2* exon 3 deletion exhibit a peculiar phenotype that includes a rare event of atrial arrhythmias and paroxysmal atrial fibrillation. The literature data showed that there was a trend in clinical events between genders: syncope and SCA were relatively higher in female patients, even at a younger age.

Therefore, detection of the deletion offers an important modality for predicting the prognosis of patients affected by LVNC with or without ventricular/atrial arrhythmias, particularly in children.

One might speculate that *RYR2* exon 3 deletion might represent a reversed phenotype for *SCN5A* with a higher percentage of affected females compared with males in *RYR2*-affected individuals.

From a practical point of view, the detection of CNVs, in genetically elusive SNVs, must be considered and can be of greater yield, especially in two major contexts: familial channelopathies and patients with extreme phenotypes, such as strong clinically suspected JLN or CPVT.

It is important to take into consideration that a comprehensive revision of the literature on patients carrying CNVs in cardiac channelopathies is still underused in different diagnostic settings. This fact leads to a limited number in the whole cohort, and subsequently, the above-mentioned observations of our cohort and those of the literature need to be validated on a multicentric level.

## 5. Conclusions

Although the presence of deletions/duplication (CNVs) in arrhythmic genes has been previously reported, this study underlines the importance of genetic testing with MLPA or a dedicated NGS pipeline for CNV identification in patients with genetically elusive routine sequencing testing. This is especially important in those with frank/extreme disease phenotypes and in familial conditions compared with sporadic ones.

We propose that screening for the genomic rearrangement of *KCNQ1*, *KCNH2*, *SCN5A*, and *RYR2* should be considered in the context of evaluating (in the routine workup/practice of) ICAs in the absence of SNVs or in non-conclusive SNVs. To increase the detection rate of cardiac channelopathies, future studies are needed to define the diagnostic protocols to be implemented in patients with ICAs, considering the relevant presence of identified CNVs. However, more CNV studies are required to establish a genotype–phenotype correlation.

## Figures and Tables

**Figure 1 biomolecules-14-01450-f001:**
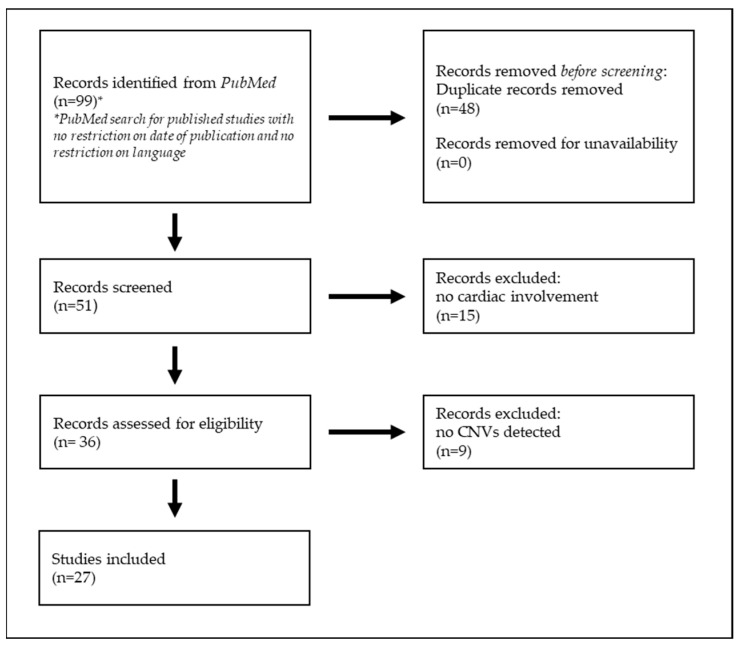
Method used to conduct the literature search for this study.

**Figure 2 biomolecules-14-01450-f002:**
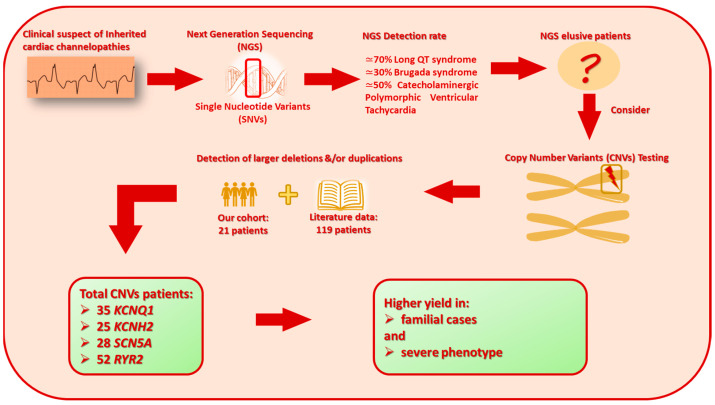
Graphical overview of current knowledge of major steps in genetic testing in patients with ICAs. The figure includes details on CNV testing, literature data, and results from this study.

**Figure 3 biomolecules-14-01450-f003:**
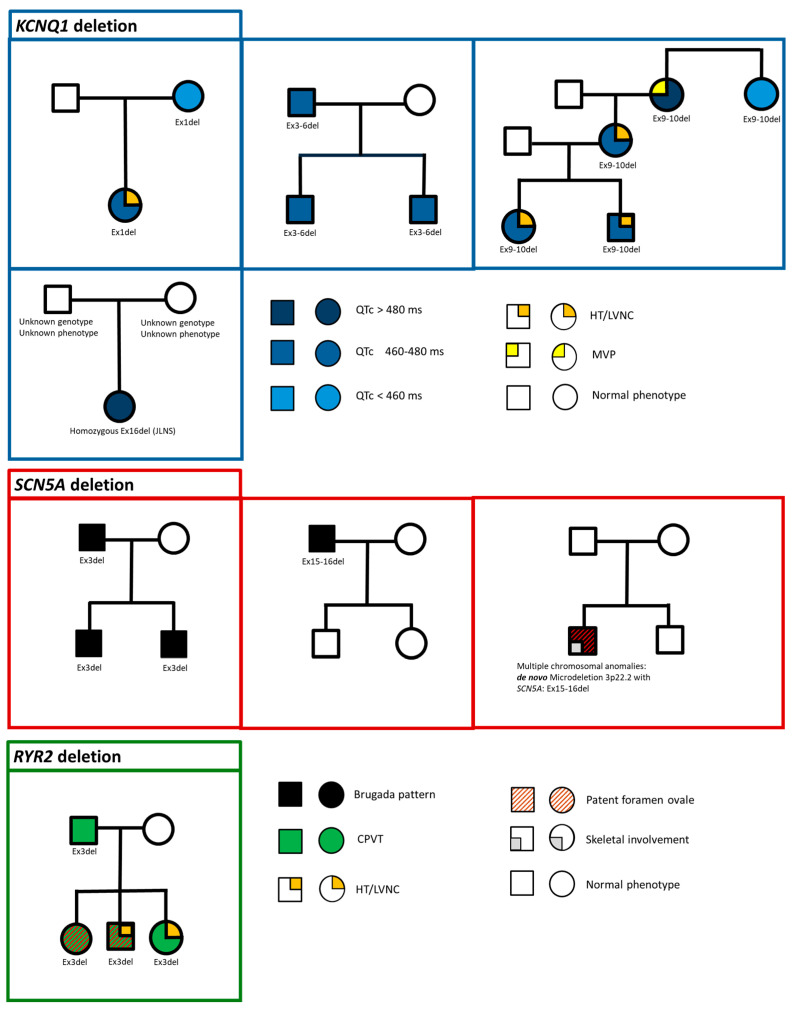
Family pedigrees of the described cohort of 21 individuals with CNV in one of the analyzed genes of cardiac channelopathies. From top to bottom: *KCNQ1*, *SCN5A*, and *RYR2* gene deletion. Phenotypic and genotypic features of probands and their relatives are shown according to the legend on the lower right of the figure. All CNVs are heterozygous unless otherwise specified. JLNS: Jervell and Lange-Nielsen syndrome. HT: hypertrabeculation. LVNC: left ventricular non-compaction. CPVT: catecholaminergic polymorphic ventricular tachycardia.

**Figure 4 biomolecules-14-01450-f004:**
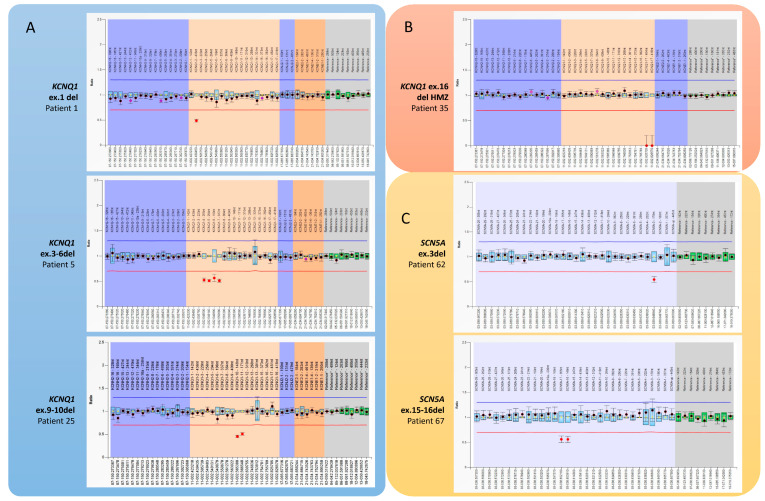
Detection of CNVs involving arrhythmic genes identified by P114 and P108 MRC Holland MLPA kits. The relative peak ratio of MLPA products in these profiles reflects the presence of exon deletions or duplications compared with reference samples. Blue/green bars represent a 95% confidence interval over the reference samples (N = 4), and dots with lines represent a 95% confidence interval estimate for each probe. The chromosomal positions and bands in the MLPA kits are all based on the hg18 genome; we convert the genomic coordinates to hg19 according to the CNVs described in the literature. For this reason, The NM_ sequence used by the company to determine a probe’s ligation site does not always correspond to the exon numbering obtained from the LRG sequences (Locus Reference Genomic). (**A**,**B**) We identified different *KCNQ1* deletions in 4 unrelated families. (**A**) Patient 1 shows a single copy loss of exon 1 in *KCNQ1* transcript variant 2 (NM_181798.1). (**B**) In patient 35, we identified a homozygous deletion of *KCNQ1* exon 16. (**C**) Deletion in *SCN5A* gene.

**Table 1 biomolecules-14-01450-t001:** (a) CNV detected in *KCNQ1*; clinical characteristics of the patients described in the literature and of our cohort. (b) CNV detected in *KCNH2*; clinical characteristics of the patients described in the literature and of our cohort. (c) CNV detected in *SCN5A*; clinical characteristics of the patients described in the literature and of our cohort. (d) CNV detected in *RYR2*; clinical characteristics of the patients described in the literature and of our cohort.

(a)											
Gene	CNV	Reference	Patient	Age at Diagnosis	Gender	QTc Interval ^‡^	Syncope	Trigger	SCA/D	Treatment	Other
** *KCNQ1* ** **NM_000218.3**	NM_181798.1 ^†^ ex.1del	This study	**1 ***	8	F	2	-	-	-	BB-nadolol	LVHT
			**2**	38	F	1	-	-	-	-	Left upper limb phocomelia
	ex.2del	Janin et al., 2021 [7]	**3 ***	-	-	-	-	-	-	-	-
	ex.3del	Tester et al., 2010 [10]	**4 ***	10	M	3	+	Exercise	-	ICD, BB-propranolol	Seizures, TdP
	ex.3-6del	This study	**5 ***	neonatal	M	2	-	-	-	BB-nadolol	-
			**6**	48	F	2	-	-	-	BB-nadolol	-
			**7**	neonatal	M	2	-	-	-	BB-nadolol	-
	ex.7del	Tester et al., 2010 [10]	**8 ***	17	F	3	+	Exercise	-	ICD, BB-nadolol	-
			**9**	13	M	3	-	-	-	BB-nadolol	-
			**10**	19	F	2	-	-	-	BB-nadolol	Abnormal T-wave
			**11**	16	M	1	-	-	-	BB-nadolol	-
			**12**	>18	F	3	-	-	-	BB-nadolol	-
	ex.7del	Janin et al., 2021 [7]	**13 ***	-	-	-	-	-	-	-	-
			**14 ***	-	-	-	-	-	-	-	-
	ex.7-8del	Barc et al., 2011 [11]	**15 ***	14	F	3	+	Stress	-	-	Broad-base T-wave
			**16**	>18	M	1	-	-	-	-	-
	ex7-8del	Campuzano et al., 2014 [13]	**17 ***	14	F	3	-	-	-	BB	-
			**18**	10	M	3	-	-	-	BB	-
			**19**	>18	F	1	-	-	-	BB	paroxysmal TC
	Ex7-9del	Hertz et al., 2015 [15]	**20**	64	F	-	-	-	+	-	-
	ex7-10del	Sung et al., 2014 [17]	**21 ***	4	M	3	+	Exercise	-	BB-atenolol	JLNS: Compound HTZ with SNV in KCNQ1, HL
			**22**	>18	M	1	-	-	-	BB	-
	ex7-10del	Sung et al., 2014 [17]	**23 ***	1	M	3	-	-	-	BB-atenolol	JLNS: Compound HTZ with SNV in KCNQ1, HL
	ex7-10del	Janin et al., 2021 [7]	**24 ***	-	-	-	-	-	-	-	-
	ex9-10del	This study	**25 ***	1	F	2	-	-	-	BB-nadolol	LVHT
			**26**	41	F	2	-	-	-	-	LVHT
			**27**	9	M	2	-	-	-	BB-nadolol	LVHT
			**28**	62	F	3	-	-	-	-	MVP
			**29**	40	F	2	-	-	-	-	-
	ex.13del	Williams et al., 2015 [14]	**30 ***	26	M	3	+	-	+	-	Obesity
	ex.13del in HMZ	Mahdieh N et al., 2020 [18]	**31 ***	7m	M	3	-	-	-	-	JLNS, HL, Seizures
	ex.13-14del	Eddy et al., 2008 [9]	**32 ***	11	M	3	+	Exercise	-	-	-
	ex16del	Zhao et al., 2023 [16]	**33 ***	20	F	3	+	-	-	BB-propranolol	JLNS: Compound HTZ with SNV in KCNQ1, HL, ID
			**34**	>18	F	-	-	-	-	-	-
	ex.16del in HMZ	This study	**35 ***	34	F	3	+	Emotional stress	-	BB-Propanolol	JLNS, VE
**(b)**											
**Gene**	**CNV**	**Reference**	**Patient**	**Age at Diagnosis**	**Gender**	**QTc Interval ^‡^**	**Syncope**	**Trigger**	**SCA/D**	**Treatment**	**Other**
** *KCNH2* ** **NM_000238.4**	ex.2dup	Stattin et al., 2012 [12]	**36 ***	-	-	-	-	-	-	-	-
	ex.4-15del	Barc et al., 2011 [11]	**37 ***	23	F	3	+	-	+	ICD	TdP, VF, bifid T-wave
			**38**	-	F	2	-	-	-	-	-
			**39**	-	M	3	-	-	-	-	-
			**40**	-	F	3	-	-	-	-	-
			**41**	-	M	3	-	-	-	-	-
			**42**	-	F	3	-	-	-	-	-
			**43**	-	F	3	-	-	-	-	-
	ex.5-11del	Janin et al., 2021 [7]	**44 ***	-	-	-	-	-	-	-	-
	ex.5-15del	Barc et al., 2011 [11]	**45 ***	28	F	3	+	Acoustic	-	-	Bifid T-wave
			**46**	-	F	2	-	-	-	-	-
			**47**	-	M	-	-	-	-	-	-
	ex.5-15del	Janin et al., 2021 [7]	**48**	-	-	-	-	-	-	-	-
	ex.6-7dup	Koopmann et al., 2006 [8]	**49 ***	17	F	3	-	-	-	BB	TdP
			**50**	-	F	-	+	Acoustic	+	BB	-
			**51**	-	M	-	-	-	-	-	-
			**52**	27	F	3	+	Dehydration	-	BB	-
			**53**	-	F	2	-	-	-	BB	-
	ex.6-14del	Eddy et al., 2008 [9]	**54 ***	22	F	3	+	Exercise/sleep	-	-	Episodes of collapse and seizures
			**55**	36	F	3	+	-	+	-	Epilepsy
	ex.9-10del	Stattin et al., 2012 [12]	**56 ***	-	-	-	-	-	-	-	-
	ex.9-14dup	Eddy et al., 2008 [9]	**57 ***	12	M	3	-	-	-	BB	-
			**58**	-	M	3	-	-	-	-	-
	Whole gene del	Janin et al., 2021 [7]	**59 ***	-	-	-	-	-	-	-	-
	Whole gene del	Janin et al., 2021 [7]	**60 ***	-	-	-	-	-	-	-	-
**(c)**											
**Gene**	**CNV**	**Reference**	**Patient**	**Age at Diagnosis**	**Gender**	**QTc Interval ^‡^**	**Syncope**	**Trigger**	**SCA/D**	**Treatment**	**Other**
** *SCN5A* ** **NM_001099404.2**	5’ upstream region	Jenewein et al., 2017 [22]	**61 ***	16	F	-	-	-	+	-	BrS
	ex.3del	This study	**62 ***	57	M	-	-	-	-	ICD	BrS
			**63**	15	M	-	-	-	-	-	BrS
			**64**	23	M	-	-	-	-	-	BrS
	ex.4del	Sonoda et al., 2018 [21]	**65 ***	25	M	-	-	-	-	ICD	VT, AVB, BrS
	ex.9-10del	Eastaugh et al., 2011 [20]	**66 ***	14	M	-	-	-	-	-	BrS, AFL
	ex.15-16del	This study	**67 ***	13	M	-	-	-	-	PM	SSS, SVT, Dysmorphic features, mild skeletal abnormalities, multiple chromosomal anomalies de novo (find text for details)
	ex.15-16del	This study	**68 ***	38	M	-	+	-	-	-	BrS
			**69**	8	F	-	-	-	-	-	Normal phenotype
	ex.17-24dup	Sonoda et al., 2018 [21]	**70 ***	11	F	-	-	-	-	-	BrS
	ex.20	Kohli et al., 2021 [24]	**71 ***	52	F	-	-	-	-	BB-flecainide catheter ablation, PM, type 2 DM, hypothyroidism	SSS, AF, BrS
			**72**	17	F	-	-	-	-	-	Normal phenotype
			**73**	16	F	-	+	Pain	-	-	BrS
			**74**	80	F	-	-	-	-	PM	BC
	ex.23del	Hertz et al., 2015 [15]	**75 ***	27	M	-	+	-	+	-	BrS
	ex.23del	Broendberg et al., 2016 [19]	**76 ***	38	M	-	+	Sleep	-	ICD	BrS
			**77**	36	F	-	-	-	-	-	Normal phenotype
			**78**	41	F	-	-	-	-	-	Normal phenotype
			**79**	66	M	-	-	-	+	-	Normal phenotype
	ex.24del	Sonoda et al., 2018 [21]	**80 ***	42	M	-	+	Exercise	+	ICD	BrS, VF
			**81**	11	F	-	-	-	-	-	Normal phenotype
	Whole gene del	Sonoda et al., 2018 [21]	**82 ***	15	M	-	+	-	-	ICD	BrS, SSS, AVB
			**83**	17	F	-	-	-	-	ICD	BrS, AVB
			**84**	10	M	-	-	-	-	-	AVB
			**85**	>18	F	-	-	-	-	-	AVB
	3p22.2del (*SCN5A*del and *SCN10A*del)	Trujillo-Quintero et al., 2019 [23]	**86 ***	13	M	-	-	-	-	ICD	BrS, AFL, SSS
			**87**	>18	M	-	-	-	-	ICD	BrS
			**88**	11	M	-	-	-	-	-	BrS
**(d)**											
**Gene**	**CNV**	**Reference**	**Patient**	**Age at Diagnosis**	**Gender**	**QTc Interval ^‡^**	**Syncope**	**Trigger**	**SCA/D**	**Treatment**	**Other**
** *RYR2* ** **NM_001035.3**	ex.3del	Bhuiyan el al., 2007 [25]	**89 ***	13	F	-	+	Exercise	-	ICD, BB-enalapril	CPVT, AVB, BC, AA(PAF), SAD, VA, DCMP
			**90**	28	F	-	+	-	+ (30y)	BB	AVB, AA(PAF), SAD, VA, DCMP
			**91**	45	M	-	-	-	-	BB	AA(PAF), SAD, VA
			**92**	>18	M	-	-	-	-	BB	VA
			**93**	>18	F	-	-	-	-	BB	AA, SAD, VA
			**94**	>18	M	-	-	-	-	BB	AA, VA
			**95**	>18	M	-	-	-	-	BB	AA, SAD, VA
			**96**	-	F	-	-	-	-	BB	AA, SAD, VA
			**97**	-	F	-	-	-	-	BB	AA, VA
			**98**	-	M	-	-	-	-	BB	AA, VA
			**99**	13	M	-	-	-	-	BB	AVB, AA(PAF), SAD, VA, DCMP
	ex.3del	Bhuiyan el al., 2007 [25]	**100 ***	45	F	-	-	Exercise	-	-	AA, VA, DCMP
			**101**	48	M	-	-	-	-	PM	SSS, AVB, AA(PAF), SAD, VA, DCMP
	ex.3del	Marjamaa et al., 2009 [26]	**102 ***	33	M	-	+	Exercise	-	-	CPVT
			**103**	-	F	-	+	Exercise	-	-	CPVT, PAF, BC, AoDil
			**104**	-	M	-	+	Exercise	-	-	CPVT, PAF
			**105**	-	F	-	+	Exercise	-	-	CPVT, BC
			**106 ***	39	M	-	+	Exercise	-	-	CPVT, PAF, BC, VT, LVNC
			**107**	-	M	-	-	-	-	-	CPVT
	ex.3del	Szentpáli et al., 2013 [27]	**108 ***	39	F	-	-	-	-	PM, ICD, BB	CPVT
	ex.3del	Ohno et al., 2014 [28]	**109 ***	17	F	-	+	Exercise, emotional stress	-	BB, ICD	CPVT, LVNC, BC, SSS
			**110**	25	F	-	+	Exercise, swimming	-	BB, ICD	CPVT, LVNC, SSS
			**111**	52	F	-	-	-	-	BB-verapamil	AF
			**112**	80	F	-	-	-	-	PM	LVNC, SSS, BC
			**113**	3	F	-	-	-	-	-	LVNC
			**114**	1	M	-	-	-	-	-	LVNC
	ex.3del	Ohno et al., 2014 [28]	**115 ***	9	F	-	+	emotional stress	-	PM, BB-carvediol	CPVT, LVNC, T-wave inversion, AVB, BC
			**116**	38	F	-	+	Exercise	-	-	CPVT, LVNC, T-wave inversion, BC
	ex.3del	Campbell et al., 2015 [13]	**117 ***	13	F	-	+	Exercise	+	ICD, BB-Nadolol, Flecainide, sympathectomy	CPVT, AF, VF, LVNC
	ex.3del	Leong et al., 2015 [29]	**118 ***	43	F	-	+	Exercise	-	BB, ICD, Ethosuximide	CPVT, epilepsy, ADHD
			**119**	-	F	-	-	-	-	BB, LCSD	CPVT, seizures
	ex.3del	Kohli et al., 2019 [31]	**120 ***	17	M	-	+	-	-	BB-nadolol(non tolerated), BB-flecainide, Sympathectomy, ICD	CPVT
	ex.3del	Dharmawan et al., 2019 [32]	**121 ***	40	F	-	+	Exercise	-	ICD	CPVT, BC, prominent U-waves, LVNC
			**122**	69	M	-	-	-	-	PM	prominent U-waves, SSS, AVB, AT
			**123**	70	F	-	-	-	-	PM	prominent U-waves, SSS
			**124**	11	F	-	-	-	-	-	LVNC, inverted T-waves
	ex.3del	Theis et al., 2021 [34]	**125 ***	2 months	M	-	-	-	-	mitral and aortic atresia, PM, sympathectomy, atrial ablation, BB-tenolol, enalapril, diuretics, tadalafil,	CPVT, HLHS, SSS, IEAT
			**126**	>18	M	-	-	-	-	ICD	CPVT, LVNC
	ex.3del	This study	**127 ***	9	F	1–2	-	-	-	BB-nadolol	CPVT, LVNC, BC, PFO
			**128**	35	M	1	-	-	-	PM, BB-Bisoprolol	CPVT, AVB-III
			**129**	15	M	1	-	-	-	PM, BB-nadolol	CPVT, LVNC (EF 50%), BC, PFO closed percutaneously
			**130**	22	F	1	-	-	-	BB-nadolol	CPVT, LVNC, BC
	5′UTR/promoter region and ex.1-4 dup in HMZ	Tester et al., 2020 [30]	**131 ***	12	F	-	-	-	+	-	-
			**132**	10	F	-	-	-	+	-	-
			**133**	3	F	-	-	-	+	-	-
			**134**	9	F	-	-	-	+	-	-
			**135 ***	-	F	-	-	-	-	-	-
			**136**	-	F	-	-	-	-	-	-
			**137**	-	F	-	-	-	+	ICD	-
			**138 ***	-	M	-	-	-	+	ICD	-
			**139**	-	M	-	+	-	-	-	-
			**140 ***	-	F	-	+	-	-	-	-

^‡^ QTc: 3: >480 ms; 2: 460–480 ms; 1: <460 ms (corrected QT interval calculated by Bazett formula); ^†^ non-canonical isoform; * proband in that specific family; AA: Atrial Arrythmia; ADHD: Attention Deficit Hyperactivity Disorder; AF: Atrial Fibrillation; AFL: Atrial Flutter; AoDil: Aortic dilatation; AT: Atrial tachycardia; AVB: Atrioventricular Block; BB: Beta-Blockers; BC: Bradycardia; BrS: Brugada syndrome; CNV: Copy number variation; CPVT: Catecholaminergic Polymorphic Ventricular Tachycardia; DCMP: Dilated cardiomyopathy; Del: Deletion; DM: Diabetes mellitus; Dup: Duplication; F: Female; HL: Hearing loss; HLHS: Hypoplastic Left Heart Syndrome; HMZ: Homozygous; HTZ: Heterozygous; ICD: Implantable Cardioverter Defibrillation; ID: Intellectual Disability; IEAT: Intractable ectopic atrial tachycardia; JLNS: Jervell and Lange Nielsen Syndrome; LVHT: Left ventricular hypertrabeculation; M: male; m: month; MVP: Mitral valve prolapse; LCSD: Left cardiac sympathetic denervation; LVNC: Left ventricular Non Compaction; PAF: Paroxysmal atrial fibrillation; PFO: Patent Foramen Ovalis; PM: Pacemaker; SAD: Sinoatrial Dysfunction; SCA/D: Sudden cardiac arrest/death; SNV: Single nucleotide variant; SSS: Sick Sinus Syndrome; SVT: Supraventricular tachycardia; TC: Tachycardia; TdP: Torsades de Pointes; VA: Ventricular Arrythmia; VE: Ventricular Extrasystoles; VF: Ventricular Fibrillation; VT: Ventricular Tachycardia.

## Data Availability

The data that support the findings of this study are available from the corresponding author upon reasonable request.

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
