# Peer review of "Copy Number Variants in Cardiac Channelopathies: Still a Missed Part in Routine Arrhythmic Diagnostics"

_biomolecules, 2024, doi:10.3390/biom14111450_

Round 1

Reviewer 1 Report

Comments and Suggestions for Authors

This review focuses on sudden cardiac death (SCD) in young caused by Inherited cardiac channelopathies. Copy Number Variants in Cardiac Channelopathies: A Potentially Missed Part in Routine Arrhythmic Diagnostics. A Multi-3 centric Experience and Literature Revision. I have the following comments:

1.        It needs to be clarified how the authors have envisioned the identification of CNV through literature revision. I guess the authors performed a literature survey, which they referred to as “literature revision,” and based on that, they generated Figure 3. The authors are suggested to present Figure 3 as Figure 1 to describe how they proceeded with a literature survey to show the current status of CNVs in genes involved in cardiac channelopathies.

2.        The authors described that young patients with CNVs in KCNQ1 show a Long QT (LQT) phenotype>480ms and a higher frequency of syncope. However, there is no statistical backup of this association, which means perhaps it is a mere coincidence unless aided by statistical analysis.

3.        The authors should perform a meta-analysis of the CNVs identified in cardiac channelopathy genes by different research groups found in the literature, including one from their own cohort. This is necessary to support the authors’ concluding remarks where they proposed that screening for genomic rearrangement of KCNQ1, KCNH2, SCN5A, 314, and RYR2 should be considered as part of routine workup/practice evaluation. Else, the authors should add a paragraph describing what more needs to be done by the researcher of the field to make CNV prevalence in cardiac channelopathy genes as part of the future diagnostic or prognostic framework in clinical decision-making.

4. The current title is too long and can be shortened to make it more concise.

Author Response

Kindly find point by point reply in attached file. 

Many thanks for your time in helping us to ameliorate our paper. 

Best

Anwar

Reviewer 2 Report

Comments and Suggestions for Authors

Start by establishing why inherited cardiac channelopathies and CNVs are significant for SCD, setting a more defined context. For instance, "Inherited cardiac channelopathies contribute significantly to sudden cardiac death (SCD) in young people."

 Clarify the study's objective more explicitly. This will help readers understand the purpose of your work from the outset.

 The results could benefit from clearer segmentation. Presenting data for each gene (KCNQ1, KCNH2, etc.) in a more uniform format or in bullet points could help readers follow the findings.

Particularly, t he passage is generally clear but could benefit from minor adjustments to improve flow and readability. The scientific terms are accurate, but a few sentences would be easier to follow with some rewording.

Rare CNV identified by MLPA and/or CMA should estimated, about genotype-phenotype correlation. 

 It should be verified whether each patient's symptoms can be fully explained by the detected CNVs.

It should be determined whether the size difference of the CNV affects the severity of the patient's symptoms.

Author Response

(The authors gave the same response as above.)

Round 2

Reviewer 1 Report

Comments and Suggestions for Authors

The revised version of manuscript is improved and acceptable for publication. 

Reviewer 2 Report

Comments and Suggestions for Authors

.